# Some Theoretical and Experimental Extensions Based on the Properties of the Intrinsic Transfer Matrix

**DOI:** 10.3390/ma15020519

**Published:** 2022-01-10

**Authors:** Nicolae Cretu, Mihail-Ioan Pop, Hank Steve Andia Prado

**Affiliations:** IEFA/Physics Department, Transilvania University of Brasov, 500036 Brasov, Romania; mihailp@unitbv.ro (M.-I.P.); hank.andia@unitbv.ro (H.S.A.P.)

**Keywords:** intrinsic transfer matrix, eigenvalues and eigenmodes, phase velocity, elastic constants

## Abstract

The work approaches new theoretical and experimental studies in the elastic characterization of materials, based on the properties of the intrinsic transfer matrix. The term ‘intrinsic transfer matrix’ was firstly introduced by us in order to characterize the system in standing wave case, when the stationary wave is confined inside the sample. An important property of the intrinsic transfer matrix is that at resonance, and in absence of attenuation, the eigenvalues are real. This property underlies a numerical method which permits to find the phase velocity for the longitudinal wave in a sample. This modal approach is a numerical method which takes into account the eigenvalues, which are analytically estimated for simple elastic systems. Such elastic systems are characterized by a simple distribution of eigenmodes, which may be easily highlighted by experiment. The paper generalizes the intrinsic transfer matrix method by including the attenuation and a study of the influence of inhomogeneity. The condition for real eigenvalues in that case shows that the frequencies of eigenmodes are not affected by attenuation. For the influence of inhomogeneity, we consider a case when the sound speed is varying along the layer’s length in the medium of interest, with an accompanying dispersion. The paper also studies the accuracy of the method in estimating the wave velocity and determines an optimal experimental setup in order to reduce the influence of frequency errors.

## 1. Introduction

A transfer matrix is an important tool in the study of wave and pulse propagation in finite and infinite homogeneous and inhomogeneous elastic media. The method is widely used in computer simulation or in elastic characterization of different kinds of elastic media [1,2,3].

The main characteristic of matrix methods with respect to other approaches is their simple and compact analytical form and the ease of application in obtaining theoretical results [4]. This has further advantages in the development of numerical methods, where a compact encoding of wave propagation and scattering allows building clear and efficient simulations, which are also easy to test and modify. Computational requirements are also relatively low, and the approach is easy to extend. The downside of matrix methods is that, while they are well adapted to 1D wave propagation, for larger dimensions they require complicated transfer matrices which are harder to work with and interpret in terms of involved wave phenomena. Nevertheless, in many applications and experimental setups, considering a 1D-wave propagation is sufficient, which allows a refined and straightforward approach by a suitable matrix method in order to simulate, explain and characterize the considered system [5].

The elements of the transfer matrix are obtained as a consequence of the boundary conditions and propagation mechanism [6,7]. For the 1D propagation and solid elastic media, boundary conditions imply continuity of the stress and of the wave function at the interface [8,9,10].

Especially in the study of multilayered media, the transfer matrix method is used in sound insulation or transmission loss factor in order to evaluate sound attenuation [11,12,13], in automotive design of the interior of the car [1,4] or the design of multiple connected mufflers [14,15]. A large volume of research refers to sonic crystals’ behavior and optimization using the transfer matrix method [16,17,18]. By using the transfer matrix, formalism is possible to model the behavior of elastic media with inhomogeneities. In that case, the transfer matrix approach combines with the finite element method in order to describe homogeneous and non-homogeneous acoustic absorbing materials. The characterized acoustic materials are mainly metamaterials made of multiple layers, where at least one layer consists of a non-homogeneous material. The equivalent transfer matrix of the non-homogeneous material is determined and, by using the equivalent transfer matrix of the nonhomogeneous material coupled analytically in a series with other transfer matrices, complex multilayer systems can be modeled easily and quickly in configurations wherein the use of finite element calculation alone will be more expensive and time consuming [19]. Other methods have been developed to numerically simulate waves in complex materials, some of which take advantage of matrix formulations [20,21,22,23]. Especially in the case of polycrystalline materials, velocity surfaces [24] or slowness surfaces [25,26] are used to characterize local anisotropy. To take into account inhomogeneity and anisotropy is important in sound wave propagation [27,28] as it gives a clearer view on wave propagation and scattering and also on the emerging sound dispersion in composite or multilayered structures.

Intrinsic transfer matrix represents a special kind of transfer matrix written for amplitudes of the Fourier components of the waves confined in an embedded elastic system. Combining the properties of the intrinsic transfer matrix with a corresponding modal analysis, we can determine some elastic constants of the system. This application of the method is presented in Section 2. By imposing the condition that the eigenvalues of the intrinsic transfer matrix are real at resonance, we can generate another form of the resonance condition for the considered elastic system. This extension is presented in Section 3.1. The resonance condition may be generalized if we consider the attenuation of the amplitudes of the involved waves. The obtained results in case of attenuation were compared with those obtained by using a simulation based on the model of the coupled oscillators (this method of simulation is very often used for the study of multilayered materials). This is presented in Section 3.2. By using computer simulation, the intrinsic transfer matrix was also applied to study the behavior of a multilayered medium with inhomogeneities, by considering the case in which one layer consists of a non-homogeneous material. This is presented in Section 3.3. One possible effect of inhomogeneity is the appearance of dispersion, i.e., frequency-dependent sound velocity. This is applied in the case of multilayered media, and also for polycrystalline materials composed of anisotropic grains such as metals, or for metamaterials in a larger sense. The simulations are presented in Section 3.3. In Section 3.4, we approach an optimization procedure for the design of a multilayer medium, where the purpose is to place the sample of interest such as to minimize the sound velocity errors due to frequency determination. The case of a ternary elastic system is being analyzed, but the study may be generalized to other complex multilayered media.

## 2. The Idea of an Intrinsic Transfer Matrix

If we consider a single elastic homogeneous layer with width l placed between two semi-infinite elastic media, the transfer matrix in the Fourier space that describes the propagation of a progressive and a regressive wave with respective amplitudes A, B is obtained as:(1)$$\left(\begin{array}{c} A_{out}\left(\omega ,l\right)\\ B_{out}\left(\omega ,l\right) \end{array}\right)=\frac{1}{4}\left(\begin{array}{cc} 1+\frac{Z}{Z_{out}} & 1-\frac{Z}{Z_{out}}\\ 1-\frac{Z}{Z_{out}} & 1+\frac{Z}{Z_{out}} \end{array}\right)\left(\begin{array}{cc} e^{i\frac{\omega }{c}l} & 0\\ 0 & e^{-i\frac{\omega }{c}l} \end{array}\right)\left(\begin{array}{cc} 1+\frac{Z_{in}}{Z} & 1-\frac{Z_{in}}{Z}\\ 1-\frac{Z_{in}}{Z} & 1+\frac{Z_{in}}{Z} \end{array}\right)\left(\begin{array}{c} A_{in}\left(\omega ,0\right)\\ B_{in}\left(\omega ,0\right) \end{array}\right)$$

In Formula (1) $Z_{in}$ and $Z_{out}$ are the elastic impedances of the two semi-infinite media, Z is the elastic impedance of the medium of interest, ω is the angular frequency and c the phase velocity of the wave. The transfer matrix in this simple case is a product of a propagation matrix P and two discontinuity matrices D, with general expressions:(2)$$P\left(\omega ,l\right)=\left(\begin{array}{cc} e^{ikl} & 0\\ 0 & e^{-ikl} \end{array}\right),\;D\left(z\right)=\frac{1}{2}\left(\begin{array}{cc} 1+z & 1-z\\ 1-z & 1+z \end{array}\right)$$
with k=ω/c as the wavenumber and $z=Z_{1}/Z_{2}$ as the relative impedance of media 1 and 2.

In the case of a standing wave, when the wave is confined only within the sample, the Fourier components of the waves inside the sample are determined only by the propagation matrix P(ω,l) in (2), which has two eigenvalues given by:(3)$$\lambda_{1,2}=\cos\frac{\omega l}{c}\pm i\;\sin\frac{\omega l}{c}$$

As it is known, if the standing wave consists of a superposition of its own eigenmodes, for which we have $\;\omega_{n}=\frac{n\pi c}{l},\;n=1,2\ldots \ldots$, it is obvious from (3) that in this special case the eigenvalues become real. Experiments confirm that we can extend the reasoning to complex embedded elastic systems and consider that for eigenfrequencies the eigenvalues become real. This important behavior of the eigenvalues can be used experimentally to find elastic constants of materials. The implied procedure is, to calculate the intrinsic transfer matrix and its eigenvalues, to experimentally find the frequencies of eigenmodes and to evaluate by a numerical analysis for which values of the wave velocity the eigenvalues are real.

The intrinsic transfer matrix method can become very useful in finding the longitudinal wave velocity in solid elastic media, especially for small samples that are not suitable to be measured by classical resonance methods. Such a sample is embedded in an elastic system with free ends containing the sample of interest and gauge materials. The entire system is excited using an impact hammer that generates an approximate Gaussian elastic pulse so that a standing wave is generated in the embedded system. The standing wave in the system is highlighted by a noncontact technique using a Doppler vibrometer, in our case an Ometron VQ-400A vibrometer. The analog signal proportional to the vibration velocity of the surface at the end of the sample is acquired with an acquisition board and spectrally processed using an analysis software, in our case LabView (or directly using an FFT analyzer). A simple FFT power spectrum allows for the determination of the frequencies of the system’s eigenmodes. Knowing the dimensions of the embedded components in the elastic system determined by length measurements, the mass density determined by weighting, and asserting the condition that the eigenvalues are real, we can estimate by a numerical analysis the wave velocity in the sample of interest.

As an example of the application of the intrinsic matrix method, in Figure 1 we illustrated the Fourier spectrum of the eigenmodes for a system consisting of two aluminum rods as gauge materials and a sample of interest of wood. The two spectra (red and black) refer to the same wood species but the two samples were cut differently, one along the fiber and the other one radially.

In Figure 2, we plotted the theoretical values of eigenfrequencies obtained from the condition that the eigenvalues are real for the ternary system illustrated in Figure 1, which consists of two aluminum rods as gauge materials $\left(l_{1}=150\;\mathrm{mm},\;l_{3}=300\;\mathrm{mm}\right)$ and the radially cut wood sample $\left(l_{2}=40.4\;\mathrm{mm}\right)$.

There is a good concordance between theory and experiment, especially for the first eigenmode of the embedded system, and as a result this recommends it in practical applications for determining the velocity of longitudinal elastic waves in small samples unsuitable for classical resonance measurements. For applications, it is important for the design of the experimental setup to be correlated to the possibility of the measuring system and the desired accuracy.

## 3. Practical Consequences of Applying the Intrinsic Transfer Matrix Method

### 3.1. A New Form of Resonance Condition

A multilayer medium with layers 1, 2, …, *n* has the intrinsic transfer matrix $T=P_{N}D_{N,N-1}\ldots P_{2}D_{1,2}P_{1}$. The propagation of a progressive and a regressive wave with amplitudes A=A(ω), B=B(ω) through the medium is generally described by:(4)$$\left(\begin{array}{c} A_{out}\\ B_{out} \end{array}\right)=T\left(\begin{array}{c} A_{in}\\ B_{in} \end{array}\right)=\left(\begin{array}{cc} a & b^{*}\\ b & a^{*} \end{array}\right)\left(\begin{array}{c} A_{in}\\ B_{in} \end{array}\right)\;$$
where a, b are complex-valued coefficients depending on ω and the layer properties and $a^{*},\;b^{*}$ are their complex conjugates. If the medium has free ends, then at resonance $A_{in}=B_{in},\;A_{out}=B_{out}$ and (4) becomes an eigenvalue equation:(5)$$\left(\begin{array}{cc} a & b^{*}\\ b & a^{*} \end{array}\right)\left(\begin{array}{c} 1\\ 1 \end{array}\right)=\lambda \left(\begin{array}{c} 1\\ 1 \end{array}\right)$$

Then, at resonance λ is real-valued and from (5) the resonance condition is:(6)Im(a)=Im(b)

Generally, (6) is a nonlinear equation in ω and it must be solved numerically in order to obtain resonance frequencies $\omega_{res}$, or the layers’ parameters if $\omega_{res}$ is known.

### 3.2. Intrinsic Transfer Matrix in Case of Attenuation

To study the influence of attenuation it is necessary to introduce attenuation coefficients of the implied materials. By considering a ternary system, where $\beta_{1},\beta_{2},\beta_{3}$ are the attenuation coefficients of amplitude and $c_{1},\;c_{2},\;c_{3}$ are the corresponding wave velocities, the intrinsic transfer matrix in the presence of attenuation will be:(7)$$TM\left(\omega \right)=\frac{1}{4}\cdot \left(\begin{array}{cc} e^{i\frac{\omega }{c_{3}}l_{3}-\beta_{3}l_{3}} & 0\\ 0 & e^{-i\frac{\omega }{c_{3}}l_{3}+\beta_{3}l_{3}} \end{array}\right)\cdot \left(\begin{array}{cc} 1+\frac{Z_{2}}{Z_{3}} & 1-\frac{Z_{2}}{Z_{3}}\\ 1-\frac{Z_{2}}{Z_{3}} & 1+\frac{Z_{2}}{Z_{3}} \end{array}\right)\cdot \left(\begin{array}{cc} e^{i\frac{\omega }{c_{2}}l_{2}-\beta_{2}l_{2}} & 0\\ 0 & e^{-i\frac{\omega }{c_{2}}l_{2}+\beta_{2}l_{2}} \end{array}\right)\cdot \left(\begin{array}{cc} 1+\frac{Z_{1}}{Z_{2}} & 1-\frac{Z_{1}}{Z_{2}}\\ 1-\frac{Z_{1}}{Z_{2}} & 1+\frac{Z_{1}}{Z_{2}} \end{array}\right)\cdot \left(\begin{array}{cc} e^{i\frac{\omega }{c_{1}}l_{1}-\beta_{1}l_{1}} & 0\\ 0 & e^{-i\frac{\omega }{c_{1}}l_{1}+\beta_{1}l_{1}} \end{array}\right)$$

The condition for real eigenvalues shows that the frequencies of eigenmodes are not affected by attenuation. The equations show that the most probable condition to have real eigenvalues is:$$\sin\left(\frac{\omega }{c_{1}}l_{1}+\frac{\omega }{c_{2}}l_{2}+\frac{\omega }{c_{1}}l_{3}\right)=0$$
$$\sin\left(\frac{\omega }{c_{1}}l_{1}-\frac{\omega }{c_{2}}l_{2}+\frac{\omega }{c_{1}}l_{3}\right)=0$$
and consequently:(8)$$\frac{\frac{l_{1}+l_{3}}{c_{1}}+\frac{l_{2}}{c_{2}}}{\frac{l_{1}+l_{3}}{c_{1}}-\frac{l_{2}}{c_{2}}}=\frac{n}{m}$$
where n and m are integers.

A numerical simulation by the coupled oscillators method confirms the independence of eigenfrequencies of the attenuation coefficients. For this, a ternary brass-aluminum-brass system with free ends was modeled as an ensemble of 5000 elements with mass connected by springs and a short impulse was applied to one end. A Fourier transform was applied to the signal at the ends. Figure 3 shows the eigenfrequencies without and with attenuation for the ternary system.

### 3.3. Influence of the Inhomogeneity Studied by Intrinsic Transfer Matrix

A special application refers to an inhomogeneous medium. Based on the intrinsic transfer matrix for a ternary system, we considered the sample of interest with a randomly varying sound speed $c_{2}$ along the middle layer, which was divided into 100 uniform slices. We numerically generated profiles for $c_{2}$ by two methods: as random values with a Gaussian distribution, or as a random walk along the layer (Figure 4).

The first method may model fractures and inclusions into the medium, while the second method generates continuous profiles and can model the influence of time-varying manufacturing processes on the material properties. Each generated profile $c_{2}\left(x\right)$ was normalized such that its mean is $\langle c_{2}\rangle =5018\;\text{m/s}$. Profiles with different value spreads, as estimated by the mean square root error $MSE\left(c_{2}\right)=\sigma \left(c_{2}\right)$, were obtained and the first eigenfrequency $f_{1}$ was computed from the intrinsic transfer matrix condition Im(λ)=0. Figure 5 shows the influence of the inhomogeneity on $f_{1}$ for different $MSE\left(c_{2}\right)$.

An exhaustive analysis of data from Figure 5 reveals that:(a)In the case of random values of $c_{2}$, taking into account 1877 profiles, we obtained a fitting function: $f_{1}=2079.3+3.93\cdot 10^{-4}\cdot MSE-4.62\cdot 10^{-6}\cdot MSE^{2}$;(b)In the case of random walk values of $c_{2}$ taking into consideration 646 profiles, the fitting function is $f_{1}=2079.3+2.66\cdot 10^{-4}\cdot MSE-4.38\cdot 10^{-6}\cdot MSE^{2}$.

It can be seen that there is a similar variation of the first resonance frequency for the two methods for $MSE\left(c_{2}\right)$ smaller than about 1000 m/s, with $f_{1}$ decreasing by about 1 Hz for $MSE\left(c_{2}\right)\;$= 500 m/s, or about 10% of $\langle c_{2}\rangle$. Thus, inhomogeneities in the sound speed have a small effect on the resonance frequency of the medium.

One possible effect of inhomogeneity is the appearance of dispersion, i.e., frequency-dependent sound velocity. This is the case for multilayer media, and also for polycrystalline materials composed of anisotropic grains, such as metals [29,30,31], or for metamaterials in a larger sense. In order to follow such an effect, we completed a series of simulations where layer 2 of the ternary brass–aluminum–brass medium is composed of such anisotropic grains and acquires a frequency-dependent sound velocity $c_{2}=c_{2}\left(f\right)$. Layers 1 and 3 were considered homogeneous and isotropic. To induce graininess, layer 2 was decomposed in $N_{F}=100$ longitudinal “fibers” (propagation directions along its length), each being composed of an equal number $N_{G}$ of grains with random lengths $l_{ij}$ having an exponential distribution [31,32]. We considered sound propagation only along the fibers, independently on each fiber. The anisotropy of grains is determined by a velocity surface $v=v_{0}\cdot F_{v}\left(\theta ,\varphi \right)$ giving the sound velocity along direction (θ,φ) in spherical coordinates. Here, $v_{0}=5018\;m$/s is the normal sound velocity in layer 2 (when considered homogneous) and $F_{v}\left(\theta ,\varphi \right)$ is a direction function; in the simplest case, this is a Fourier series in θ,φ of the form:$$F_{v}\left(\theta ,\varphi \right)=C_{v}\cdot \overset{\infty }{\underset{m=0}{\sum }}\left(A_{m}^{\theta }\sin\left(m\theta \right)+B_{m}^{\theta }\cos\left(m\theta \right)\right)\cdot \overset{\infty }{\underset{n=0}{\sum }}\left(A_{n}^{\varphi }\sin\left(n\varphi \right)+B_{n}^{\varphi }\cos\left(n\varphi \right)\right)$$
but it can have other expressions too. For example, $C_{v}$ is a constant determined by the condition:$$\langle F_{v}\rangle =\frac{1}{4\pi }\int_{\theta =0}^{\pi }\int_{\varphi =0}^{2\pi }F_{v}\left(\theta ,\varphi \right)\sin\left(\theta \right)d\theta \;d\varphi =1$$

Different other conditions can be imposed to the function $F_{v}\left(\theta ,\varphi \right)$, e.g., $F_{v}\left(\pi -\theta ,\pi +\varphi \right)=F_{v}\left(\theta ,\varphi \right)$, which ensures equal sound velocities in opposite directions. However, for metamaterials, this condition may not hold. In the simulation, for a given material in layer 2, a specific expression for $F_{v}\left(\theta ,\varphi \right)$ was applied to all the grains and fibers; for each grain, the direction of propagation was chosen at random, as $\left(\theta ,\varphi \right)=\left(\mathrm{acos}\left(1-2r_{1}\right),2\pi r_{2}\right)$, where $r_{1},r_{2}$ are uniformly distributed random numbers between 0 and 1. Four velocity surfaces were used in (Figure 6), given by:(9a)$$F_{v}\left(\theta ,\varphi \right)=1+0.05\;\cos\left(2\theta \right)$$
(9b)$$F_{v}\left(\theta ,\varphi \right)=1+0.05\;\cos\left(4\theta \right)$$
(9c)$$F_{v}\left(\theta ,\varphi \right)=1+\left(0.05\;\cos\left(4\theta \right)-0.025\;\cos\left(8\theta \right)\right)\cdot \cos\left(4\varphi \right);$$
(9d)$$F_{v}\left(\theta ,\varphi \right)=\left(1-\left(\pi -\theta \right)e^{\theta -\pi }\right)\cdot \left(1+\left(0.05\;\cos\left(4\theta \right)-0.025\;\cos\left(8\theta \right)\right)\cdot \cos\left(4\varphi \right)\right).$$

Surface (a) is suitable for modeling metals with cubic crystals [30]. Surface (d) in particular is highly asymmetric and could be achieved in a metamaterial.

At the input of layer 2 a pair of waves was applied with $A_{in}=1,\;B_{in}=0$ for a range of frequencies. The outputs for all fibers were averaged. An effective sound velocity for layer 2 $c_{2}{}_{eff}$ was computed from the average output: $\langle A_{2\;out}\rangle =\left|\langle A_{2\;out}\rangle \right|\exp\left(i\frac{\omega }{c_{2}{}_{eff}}l_{2}\right)$. Examples of effective velocities for multigrain materials can be seen in Figure 7. Additionally, resonance frequencies for the whole ternary medium were obtained and their variation with the grain number $N_{G}$ was studied (Figure 8).

Anisotropy induces a frequency-dependent sound velocity (Figure 7), and this effect is stronger for a small number of grains (large grain size), while for small grains the effective velocity is almost constant. The average effective velocity is close to that of the homogeneous layer, except for the velocity surface (d), which induces a strong anisotropy.

The considered dispersion has a minor effect on the resonances of the ternary medium (Figure 8). This is partly due to the fact that layer 2 is much shorter than layers 1 and 3, but also to the fact that the considered surface velocities are close to the isotropic one. The relative variation of the resonance frequency is quite small (around $10^{-4}$, or 0.2 Hz for the first resonance), and it is smaller for large resonance frequencies. A relatively large change in the resonance frequency occurs for the velocity surface (d) (close to $10^{-3}$, or 2 Hz for the first resonance). The case where layers 1 and 3 are also anisotropic was not considered.

### 3.4. A Proposed Optimization Alghorithm Based on the Intrinsic Transfer Matrix

An optimization procedure for the design of a multilayer medium is described, where the purpose is to place the measured sample such as to minimize the sound velocity errors due to frequency determination. Typically, this is applied to a ternary (or more complex) multilayer medium consisting of a known material 1 and the sample material 2. The density is known for both; the sound velocity is known for material 1 only. The first *N* resonance frequencies $\omega_{i}$ obey a resonance condition (6) of the general form:(10)$$C\left(c_{2},\omega_{i}\right)=0,\;i=1,2,\ldots ,\;N$$

For example, for a binary medium with material order 1–2 and a ternary medium 1–2–1, with layer lengths $l_{i}$, $k_{i}=\omega /c_{i}$ and $z=Z_{1}/Z_{2}$, (10) we have the formulas:(11)$$C_{2}:\;\left(z+1\right)\sin\left(k_{1}l_{1}+k_{2}l_{2}\right)+\left(z-1\right)\sin\left(k_{1}l_{1}-k_{2}l_{2}\right)=0$$
(12)$$C_{3}:\;{\left(z+1\right)}^{2}\sin\left(k_{1}l_{1}+k_{2}l_{2}+k_{1}l_{3}\right)-\left(z^{2}-1\right)\sin\left(k_{1}l_{1}+k_{2}l_{2}-k_{1}l_{3}\right)-{\left(z-1\right)}^{2}\sin\left(k_{1}l_{1}-k_{2}l_{2}+k_{1}l_{3}\right)+\;\left(z^{2}-1\right)\sin\left(k_{1}l_{1}-k_{2}l_{2}-k_{1}l_{3}\right)=0$$

Typically, the experimental resonances $\omega_{exp,\;i}=2\pi f_{exp,i}$, i=1,…,N will have an error due to the discrete Fourier transform used in their determination. To further estimate $c_{2}$, theoretical resonances, $\omega_{th,\;i}\left(c_{2}\right)$ are obtained from (10) which then are matched to $\omega_{exp,\;i}$ by minimizing the convex error:(13)$$E\left(c_{2}\right)=\overset{N}{\underset{i=1}{\sum }}{\left(\omega_{th,\;i}\left(c_{2}\right)-\omega_{exp,\;i}\right)}^{2}$$
(14)$$c_{2}=\mathrm{argmin}\left\{E\left(c_{2}\right)\;:\;C\left(c_{2},\omega_{th,\;i}\left(c_{2}\right)\right)=0\right\}$$

Thus, the estimated $c_{2}$ will have an error due to resonance frequency errors. The error of $c_{2}$ also depends on the position of sample material 2 in the multilayer medium, which allows one to minimize this error.

For simplicity, let us consider a ternary medium where the sample 2 must be placed at optimal position $l_{1}$. Furthermore, we assume $\omega_{exp,\;i}$ are uniformly distributed in intervals $\omega_{exp,\;i}\in \left[\omega_{th,\;i}\left(c_{2}\right)-\Delta \omega ;\omega_{th,\;i}\left(c_{2}\right)+\Delta \omega \right]$ with a fixed Δω. Then, the standard deviation of $\omega_{exp,\;i}$ is $\sigma \left(\omega_{exp,\;i}\right)=\Delta \omega /\sqrt{3}$ and assuming this is small compared to $\omega_{exp,\;i}$, the standard deviation of $c_{2}=c_{2}\left(\omega_{exp,1},\omega_{exp,2},\ldots ,\omega_{exp,N}\right)$ is estimated by error propagation:(15)$$\sigma \left(c_{2}\right)=\sqrt{\overset{N}{\underset{i=1}{\sum }}{\left(\frac{\partial c_{2}}{\partial \omega_{i}}\right)}^{2}}\cdot \sigma \left(\omega_{exp,\;i}\right)=\left\|\frac{\partial c_{2}}{\partial \omega_{i}}\right\|\cdot \frac{\Delta \omega }{\sqrt{3}}$$
where $\left\|v_{i}\right\|$ is the Euclidean norm of a vector v. Thus $l_{1}$ is given by the minimum of (15) with $c_{2}$ from (14):(16)$$l_{1}=\mathrm{argmin}\left\{\|\frac{\partial c_{2}}{\partial \omega_{i}}\|\;:\left(14\right)\right\}$$

Steps (14) and (16) can be repeated to further improve $l_{1}$; usually one pass and going through (14) again is enough to minimize (15).

Generally, estimating numerical derivatives and performing, e.g., a least squares minimization in (14) and (16) may be time and computation intensive. Obtaining an analytical formula of $\partial c_{2}/\partial \omega_{i}$ from (10) and (16) speeds up computation. Differentiating (10) with respect to $c_{2}$ twice at a fixed i, with $\omega_{i}=\omega_{i}\left(c_{2}\right)$ one obtains:(17)dωidc2=−∂C∂c2∂C∂ωi
(18)d2ωidc22=−1∂C∂ωi[∂2C∂ωi2(dωidc2)2+2∂2C∂ωi ∂c2dωidc2+∂2C∂c22]

From (14), after labeling $\omega_{i}\left(c_{2}\right)=\omega_{th,\;i}\left(c_{2}\right)$, it follows:(19)$$\frac{dE}{dc_{2}}=0\overset{\;}{\Rightarrow }\overset{N}{\underset{i=1}{\sum }}\left(\omega_{\;i}\left(c_{2}\right)-\omega_{exp,\;i}\right){\frac{d\omega_{i}}{dc_{2}}}^{\;}=0$$

To determine $\partial c_{2}/\partial \omega_{i}$ in (15), we apply a small perturbation $\delta \omega_{exp,\;i}$ in $\omega_{exp,\;i}$ with a corresponding perturbation $\delta c_{2}$ of $c_{2}$. Then (19) becomes:$$\overset{N}{\underset{i=1}{\sum }}\left(\omega_{\;i}\left(c_{2}+\delta c_{2}\right)-\omega_{exp,\;i}-\delta \omega_{exp,\;i}\;\right)\frac{d\omega_{i}}{dc_{2}}{\left(c_{2}+\delta c_{2}\right)}^{\;}=0$$

By retaining terms up to first-order perturbations, we obtain:$$\overset{N}{\underset{i=1}{\sum }}{\left[\left(\omega_{\;i}\left(c_{2}\right)-\omega_{exp,\;i}\;\right)\frac{d^{2}\omega_{i}\left(c_{2}\right)}{dc_{2}^{2}}+{\left(\frac{d\omega_{i}\left(c_{2}\right)}{dc_{2}}\right)}^{2}\right]}^{\;}\delta c_{2}=\overset{N}{\underset{i=1}{\sum }}\frac{d\omega_{i}\left(c_{2}\right)}{dc_{2}}\delta \omega_{exp,\;i}{}^{\;}$$

For fixed k, take in the right-hand side $\delta \omega_{exp,\;i}=\delta \omega_{exp,\;k}$ if i=k; else $\delta \omega_{exp,\;i}=0$. Then:(20)$$\frac{\partial c_{2}}{\partial \omega_{k}{}_{\;}}=\frac{\delta c_{2}}{\delta \omega_{exp,\;k}}=\frac{\frac{d\omega_{k}\left(c_{2}\right)}{dc_{2}}}{\sum_{i=1}^{N}{\left[\left(\omega_{\;i}\left(c_{2}\right)-\omega_{exp,\;i}\;\right)\frac{d^{2}\omega_{i}\left(c_{2}\right)}{dc_{2}^{2}}+{\left(\frac{d\omega_{i}\left(c_{2}\right)}{dc_{2}}\right)}^{2}\right]}^{\;}}$$
(21)$$\left\|\frac{\partial c_{2}}{\partial \omega_{i}}\right\|=\sqrt{\overset{N}{\underset{i=1}{\sum }}{\left(\frac{\partial c_{2}}{\partial \omega_{i}}\right)}^{2}}=\frac{\sqrt{\sum_{i=1}^{N}{\left(\frac{d\omega_{i}\left(c_{2}\right)}{dc_{2}}\right)}^{2}}}{\left|\sum_{i=1}^{N}{\left[\left(\omega_{\;i}\left(c_{2}\right)-\omega_{exp,\;i}\;\right)\frac{d^{2}\omega_{i}\left(c_{2}\right)}{dc_{2}^{2}}+{\left(\frac{d\omega_{i}\left(c_{2}\right)}{dc_{2}}\right)}^{2}\right]}^{\;}\right|}$$
where $c_{2}$ is determined from (14) and $\omega_{\;i}=\omega_{\;th,i}\left(c_{2}\right)$ from (10). The derivatives with respect to $c_{2}$ are given by (17) and (18). In particular, if one solves (16) with just the theoretical resonance spectrum, i.e., $\omega_{exp,\;i}=\omega_{\;th,i}$ then:(22)$$\left\|\frac{\partial c_{2}}{\partial \omega_{i}}\right\|=\frac{1}{\sqrt{\sum_{i=1}^{N}{\left(\frac{d\omega_{i}\left(c_{2}\right)}{dc_{2}}\right)}^{2}}}=\frac{1}{\frac{d\omega_{i}\left(c_{2}\right)}{dc_{2}}}$$

A numerical analysis was performed to illustrate the approach. A ternary medium made of brass (layers 1 and 3) and aluminum (layer 2) was considered, with layer lengths $l_{1}=544\;\mathrm{mm}$, $l_{2}=18.16\;\mathrm{mm}$, $l_{3}=251.5\;\mathrm{mm}$ and densities $\rho_{1}=\rho_{3}=8315{\;\mathrm{kg}/m}^{3}$, $\rho_{2}=2713\;{\mathrm{kg}/m}^{3}$. The sound speed in brass is considered known $c_{1}=c_{3}=3373\;m/s$ while $c_{2}$ is to be determined from the resonance spectrum; typically, a value $c_{2}=5018\;m/s$ was obtained and is used in the analysis below. This medium has the first 4 resonances at 2104.03 Hz, 4205.93 Hz, 6303.47 Hz and 8394.18 Hz. Next, the frequency error was considered fixed Δf=10 Hz, and the optimum distance $l_{1}$ that minimizes (15) was obtained from (16). When modifying $l_{1}$ in the analysis below the total length of the medium was kept constant.

The norm (22) as a function of $l_{1}$ was of interest (Figure 9). If the sample is placed at the ends of the medium, $c_{2}$ has very large errors, while inside it lies close to the measured value $c_{2}=5018\;m/s$. The positions of minimum error for $c_{2}$ lie at about $l_{1}=10\;\mathrm{cm}$ and $l_{1}=70\;\mathrm{cm}$, but also positions at about $l_{1}=30\;\mathrm{cm}$ and $l_{1}=50\;\mathrm{cm}$ trigger similar errors. The average of $c_{2}$ obtained for different resonance spectra approximates well the known value (Figure 10), except at the ends, where $\sigma \left(c_{2}\right)$ is very large.

We obtained the dependence of (22) with respect to $c_{2}$ and $l_{2}$ (Figure 11), and the number of resonances N (Figure 12). The norm (22) decreases with decreasing velocities $c_{2}$, increasing sample lengths $l_{2}$, and with increasing N. For a fixed N, the norm has N local minima, most of which have about the same minimum value. The absolute minimum is reached at the outermost local minima, and this minimum moves closer to the ends when N increases, which could make it difficult to place the sample there. On the other hand, placing the sample at the very end would give the greatest errors for $c_{2}$. So, a binary medium is not very appropriate for sound velocity measurements.

A 5-layer medium was also investigated, with layer ordering 1–2–1–2–1, with $l_{2}=l_{4}=18.16$ mm and $l_{1}+l_{3}+l_{5}=795.5$ mm fixed. The norm (22) was minimized and optimal values for *l*_1_ and *l*_3_ were obtained (Figure 13). Thus, the optimization algorithm can be applied even for complex experimental multilayer media.

## 4. Conclusions

This work presents a method of determining the longitudinal wave velocity at a normal incidence in a sample which cannot be measured by the classical resonance method. The method suggests embedding the sample in an elastic system and then determining the frequencies of the eigenmodes of the elastic system. Taking into account that for a standing wave in the system, for such a wave, only the intrinsic matrix can be considered in a wave transfer; for simple systems it can be calculated, and we can find an analytical expression for it. Generalizing for the elastic systems the behavior of the eigenvalues of the intrinsic transfer matrix, which for eigenfrequencies become real, we can determine by a numerical analysis the wave velocity in the sample of interest and then the corresponding elastic constants. From the experimental point of view, a ternary system is preferred because such a system preserves much better the longitudinal plan wave, a special case for which the transfer matrix has a simple analytical form. An example of the application of the method is illustrated for a ternary aluminum–wood–aluminum system, which shows the sensitivity of the method.

The paper presents also some experimental and computational studies that refer to the possibilities and accuracy of the method. The method can be generalized so as to take into account the attenuation. In that case, the condition for real eigenvalues shows that the frequencies of eigenmodes are not affected by attenuation.

The case of an inhomogeneous medium is also analyzed by considering two kinds of inhomogeneities. The first can model fractures and inclusions into the medium, while the second generates continuous profiles and it can model the influence of time-varying manufacturing processes on the acoustical properties. Inhomogeneity may also induce dispersion in the medium, and this was analyzed too; a small influence on resonance frequencies was detected.

The paper also proposes an optimization procedure based on the properties of the intrinsic transfer matrix that makes it possible to place the measured sample in a multilayer medium in such a way as to minimize the errors for the measured sound velocity due to frequency errors. It is very clear that, from an experimental point of view, good estimations of the wave velocity can be obtained only by a very precise Fourier analysis; hence, a good frequency resolution will lead to a good estimation of the phase velocity of the wave. Nevertheless, precision can be increased by choosing an optimal position of the sample inside the experimental setup with respect to frequency error propagation to other measures of interest that are determined.

## Figures and Tables

**Figure 1 materials-15-00519-f001:**
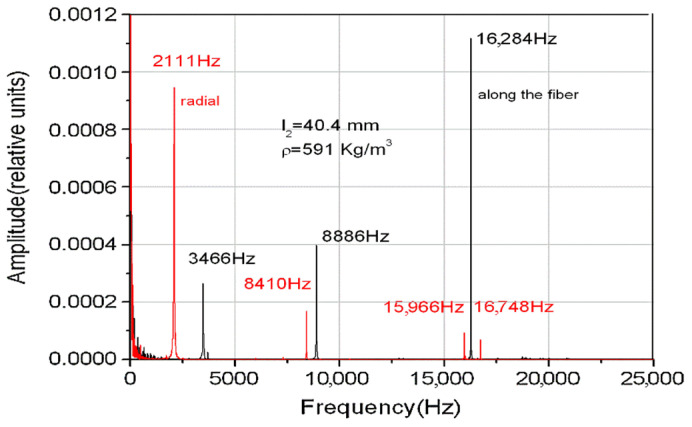
Fourier spectrum and the eigenmodes frequencies.

**Figure 2 materials-15-00519-f002:**
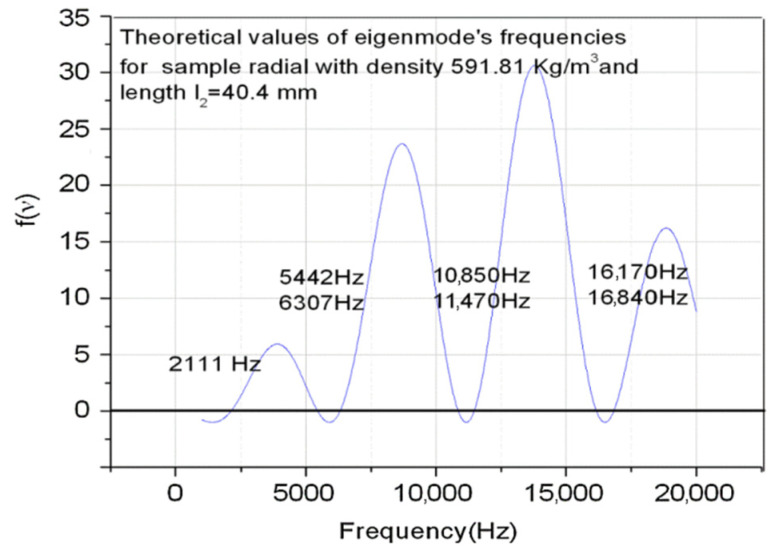
Theoretical eigenfrequencies for an aluminum-wood-aluminum system.

**Figure 3 materials-15-00519-f003:**
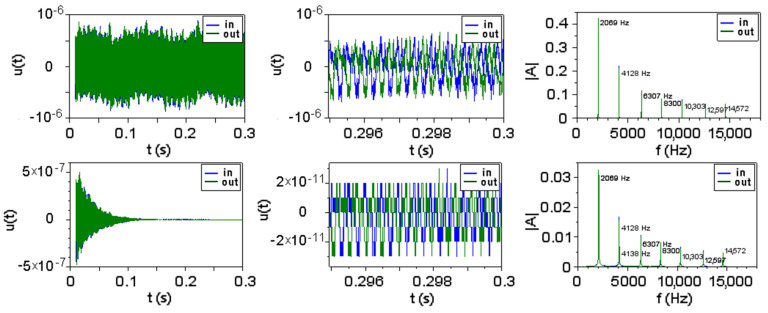
The eigenfrequencies of a brass-aluminum-brass system without (above) and with (below) attenuation.

**Figure 4 materials-15-00519-f004:**
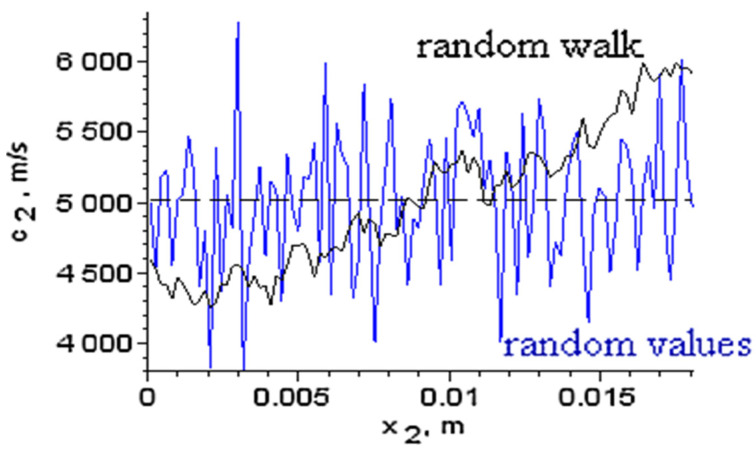
Random sound speeds generated for the medium of interest by two methods.

**Figure 5 materials-15-00519-f005:**
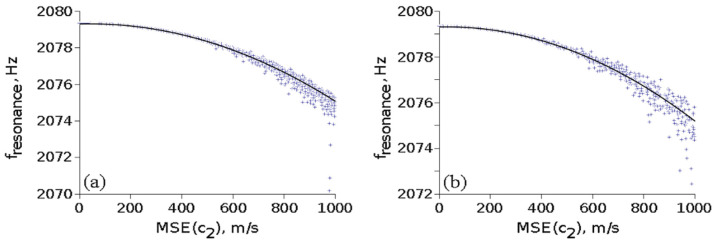
The influence of medium inhomogeneity on its first eigenfrequency for $c_{2}$ with (**a**) random values; (**b**) random walk values.

**Figure 6 materials-15-00519-f006:**
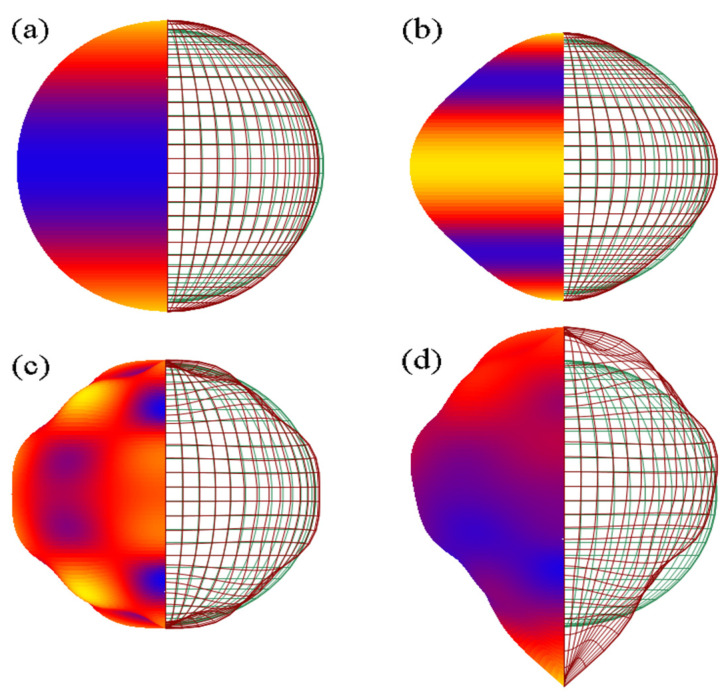
The four velocity surfaces $F_{v}\left(\theta ,\varphi \right)$ in 3D that were used to describe anisotropy in sound wave propagation. On the left, the surface colors represent the distance from the center, from small (blue) to large (yellow). On the right, the meshes represent the anisotropic (dark red) and isotropic (green) surface velocities. (**a**–**d**) correspond to Equations (9a)–(9d) respectively.

**Figure 7 materials-15-00519-f007:**
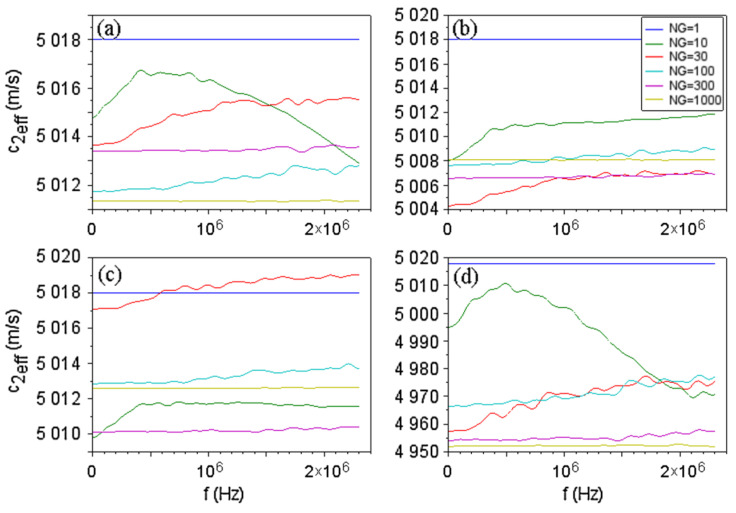
Effective sound velocities in layer 2 as the function of frequency obtained with the velocity surfaces in Figure 6, for different grain number $N_{G}$ along fibers. The case $N_{G}=1$ is for the homogeneous (monocrystalline) layer 2. The legend applies to all graphs. (**a**–**d**) correspond to Equations (9a)–(9d) respectively.

**Figure 8 materials-15-00519-f008:**
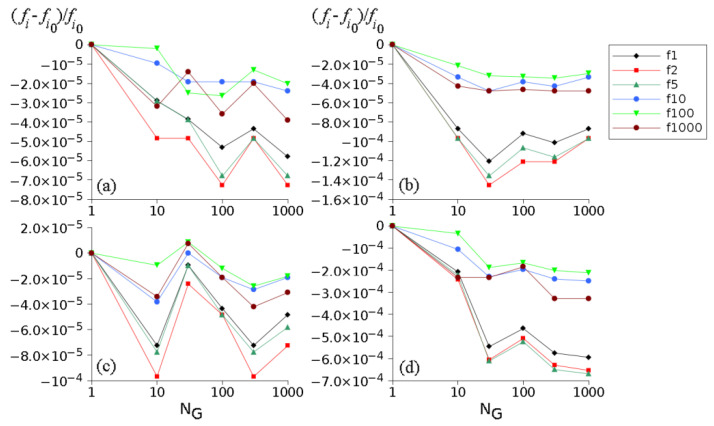
Graphs of relative variations of resonance frequencies for the ternary medium as a function of $N_{G}$ for the velocity surfaces in Figure 6 and velocity profiles in Figure 7. Here, $f_{i}{}_{0}$ and $f_{i}$ are resonance frequencies with homogeneous ($N_{G}=1$) and, respectively, grainy layer 2. The first resonance for $N_{G}=1$ is at 2068.36 Hz, while the next resonances are approximate multiples of it. The resonances No. 1,2,5,10,100 and 1000 were considered. (**a**–**d**) correspond to Equations (9a)–(9d) respectively.

**Figure 9 materials-15-00519-f009:**
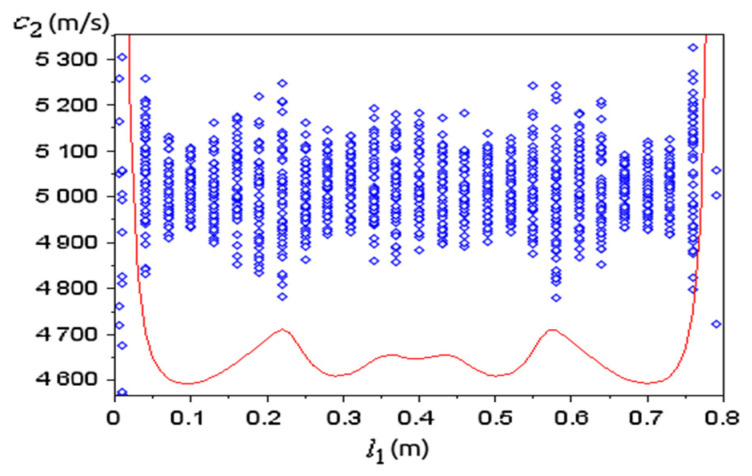
Optimum values of $c_{2}$ (blue dots) obtained from (14) for random sets of *N* = 4 resonances with frequency error Δf=10 Hz, for different positions $l_{1}$ of the sample. The red line is $\left\|\frac{\partial c_{2}}{\partial \omega_{i}}\right\|$ from (22), not to scale. The total length of the medium is constant.

**Figure 10 materials-15-00519-f010:**
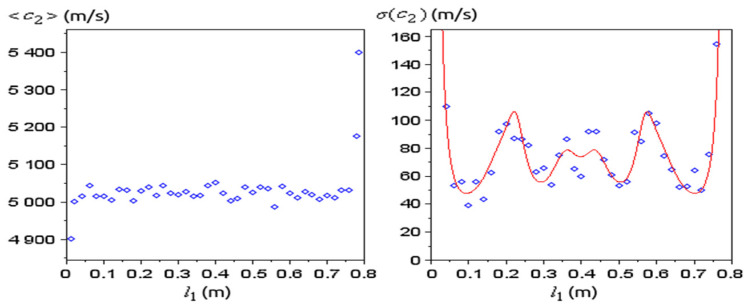
At different positions $l_{1}$, 20 sets of resonance spectra were randomly generated with the same Δf and for each set the optimum $c_{2}$ was obtained from (14) (the average of the 20 values of $c_{2}$ is plotted to the left); the standard deviation of $c_{2}$ (right) was computed numerically (blue dots) and theoretically from (15) (red curve).

**Figure 11 materials-15-00519-f011:**
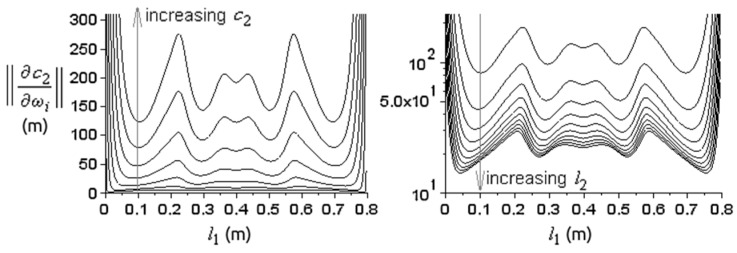
The graph of norm (22) as a function of $l_{1}$ for constant $l_{2}$ and different values of $c_{2}$ (left); and for constant $c_{2}$ and different values of $l_{2}$ (right, vertical log scale). On the left, $l_{2}$ = 18.16 mm and $c_{2}$ = 1000–6000 m/s at steps of 1000 m/s, for the same $\rho_{2}$. On the right, $c_{2}$ = 5018 m/s and $l_{2}$=1–10 cm at steps of 1 cm. The first N=4 resonances were used.

**Figure 12 materials-15-00519-f012:**
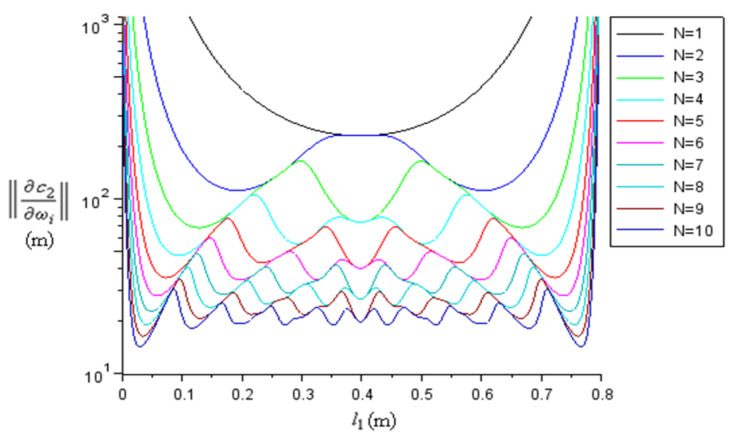
The graph of (22) (log scale) for different numbers of resonances N, with constant $c_{2}$ and $l_{2}$.

**Figure 13 materials-15-00519-f013:**
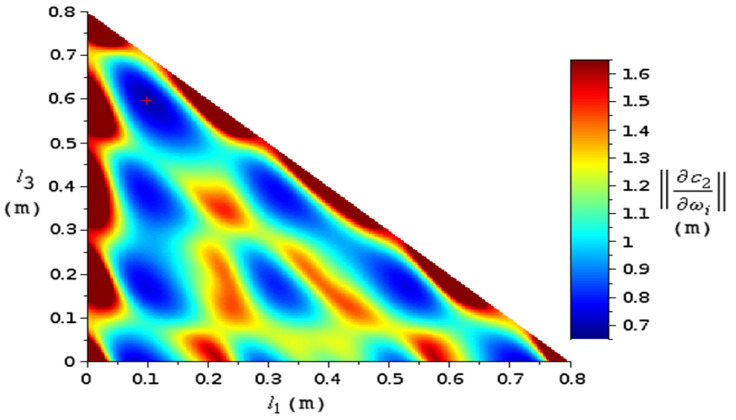
Map of the norm (22) in $\sigma \left(c_{2}\right)$ as function of $l_{1}$ and $l_{3}$ for a five-layer medium with two identical samples, with N=4 resonances. The total length was fixed. The high values are trimmed at the edges. The absolute minimum is marked with a red “plus” sign.

## Data Availability

No data was made public.

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
