# Peer review of "Some Theoretical and Experimental Extensions Based on the Properties of the Intrinsic Transfer Matrix"

_materials, 2022, doi:10.3390/ma15020519_

Round 1

Reviewer 1 Report

Title: Some theorical and experimental extensions based on the properties of the intrinsic transfer matrix

Authors: Mihail-Ioan Pop, Hank Steve Andia Prado and Nicolae Cretu

This paper studies the elastic characterization of materials based on the properties of the intrinsic transfer matrix. The results presented are interesting, however there are some points that need to be revise before the paper can be considered for publication:

The Introduction section is not detailed enough, this section is very important as it highlight the motivations and the novelties of the work. Transfer matrix is a tool that is widely used in elastic media. The authors should point out more in detail its applications, the main pros and cons of using it in each application, and from there deduce the need of using a more advanced approach.

The definition of the intrinsic transfer matrix is well presented. If you can improve the Introduction, I think that it can highlight more the motivation and the novelties of this work.

In the conclusion, before presenting the main findings, the authors should recall the motivations of the paper.

Minor points:

  • The axis numbers of the Figures need to be enhanced. The quality is very low when zoomed out, maybe put them in larger scale.
  • Some English errors need to be revised.

Author Response

Reviwer 1 Comments:

This paper studies the elastic characterization of materials based on the properties of the intrinsic transfer matrix. The results presented are interesting, however there are some points that need to be revise before the paper can be considered for publication:

  • The Introduction section is not detailed enough, this section is very important as it highlight the motivations and the novelties of the work. Transfer matrix is a tool that is widely used in elastic media. The authors should point out more in detail its applications, the main pros and cons of using it in each application, and from there deduce the need of using a more advanced approach.The definition of the intrinsic transfer matrix is well presented. If you can improve the Introduction, I think that it can highlight more the motivation and the novelties of this work.

In the conclusion, before presenting the main findings, the authors should recall the motivations of the paper.

ANSWER: We consider that the referent's observation is well-founded. Consequently, we have almost completely rewritten the introductory part of the article, including the motivation for using the intrinsic transfer matrix and ways of application and simulation. We have also inserted a more recent and significant bibliography. The new version of Introduction is:

                            ”Transfer matrix is an important tool in the study of waves and pulses propagation in finite and infinite homogeneous and inhomogeneous elastic media. The method is widely used in computer simulation or in elastic characterization of different kinds of elastic media. [1],[2],[5]

The main characteristic of matrix methods with respect to other approaches is their simple and compact analytical form and the ease of application in obtaining theoretical results. This has further advantages in the development of numerical methods, where a compact encoding of wave propagation and scattering allows building clear and efficient simulations, which are also easy to test and modify. Computational requirements are also relatively low and the approach is easy to extend. The downside of matrix methods is that while they are well adapted to 1D wave propagation, for larger dimensions they require complicated transfer matrices which are harder to work with and interpret in terms of involved wave phenomena. Nevertheless, in many applications and experimental setups considering a 1D wave propagation is sufficient, which allows an elegant and straightforward approach by a suitable matrix method in order to simulate, explain and characterise the considered system [25].

The elements of the transfer matrix are obtained as a consequence of the boundary conditions and propagation mechanism [9],[10]. For the 1D propagation and solid elastic media, boundary conditions imply continuity of the stress and of the wave function at the interface [3],[4],[12].

Especially in the study of multilayered media the transfer matrix method is used in sound insulation or transmission loss factor, in order to evaluate sound attenuation[8],[13],[14], in automotive design of the interior of the car[15] or the design of multiply connected mufflers[7],[16]. A large volume of research and studies refers to sonic crystals behavior and optimization using the transfer matrix method[18-20]. By using the transfer matrix formalism is possible to model the behavior of elastic media with inhomogeneities. In that case, transfer matrix approach combines with finite element method  in order to describe  homogeneous and non-homogeneous acoustic absorbing materials. The characterized acoustic materials are mainly metamaterials made of multiple layers, where at least one layer consists of a non-homogeneous material. The equivalent transfer matrix of the non-homogeneous material is determined and by using the equivalent transfer matrix of the nonhomogeneous material coupled analytically in series with other transfer matrices, complex multilayer systems can be modeled easily and quickly in configurations wherein the use of finite element calculation alone will be more expensive and time consumming [11]. Different other methods have been developed to numerically simulate waves in complex materials, some of which take advantage of matrix formulations [26][27]. Especially in the case of polycrystalline materials, velocity surfaces [28] or slowness surfaces [29][30] are used to characterise local anisotropy. Taking into account inhomogeneity and anisotropy is important in sound wave propagation [31][32] as it gives a clearer view on wave propagation and scattering and also on the emerging sound dispersion in composite or multilayered structures.

                               Intrinsic transfer matrix represents a special kind of transfer matrix written for amplitudes of the Fourier components of the waves confined in an embedded elastical system. Combining the properties of the intrinsic transfer matrix with a corresponding modal analysis, we can determine some elastic constants of the system. This application of the method is presented in section 2. By imposing the condition that at resonance the eigenvalues of the intrinsic transfer matrix are real, we can generate another form of the resonance condition for the considered elastic system. This extension is presented in section 3.1. The resonance condition may be generalized if we consider the attenuation of the amplitudes of the involved waves. The obtained results in case of attenuation were compared with those obtained by using a simulation based on the model of the coupled oscillators (this method of simulation is very often used for study of multilayered materials). This is presented in section 3.2.  The intrinsic transfer matrix was also applied to study by computer simulation the behavior of a multilayered medium with inhomogeneities, by considering the case in which one layer consists of a nonhomogeneous material. This is presented in section 3.3. One possible effect of inhomogeneity is the appearance of dispersion, i.e. frequency-dependent sound velocity. This is applied in the case of multilayered media, and also for polycrystalline materials composed of anisotropic grains such as metals, or for metamaterials in a larger sense. The simulations are presented in section 3.3. In section 3.4 we approach an optimization procedure for the design of a multilayered medium, where the purpose is to place the sample of interest such as to minimize the sound velocity errors due to frequency determination. The case of a ternary elastic system is being analysed, but the study may be generalized to other complex multilayered media.”

Minor points:

  • The axis numbers of the Figures need to be enhanced. The quality is very low when zoomed out, maybe put them in larger scale.

ANSWER: All figures have been redrawn so that the values of the physical meanings involved are visible and have also been converted into a format agreed by the publisher.

  • Some English errors need to be revised.

ANSWER: The text has been checked again using the spelling option.

Reviewer 2 Report

I have reviewed the article titled “Some theoretical and experimental extensions based on the properties of the intrinsic transfer matrix.” The article is very well written. However, as a reviewer, I have noticed that the novelty is not highlighted in this work. The technique transfer matrix method is quite old, only modified with the approach used in this area of research.

The experimental section and the valid reference must be quoted with all the test specifications, conditions, and standards.

Material orientation is essential, and there are plenty of known reasons and equation conditions available.

The influence of in-homogeneity details was not discussed in the existing literature.

Some of the references used in the studies are very old. Kindly include the new articles published in the last 5 years.

Author Response

ANSWERS to REVIEWER 2 COMMENTS

  • I have reviewed the article titled “Some theoretical and experimental extensions based on the properties of the intrinsic transfer matrix.” The article is very well written. However, as a reviewer, I have noticed that the novelty is not highlighted in this work. The technique transfer matrix method is quite old, only modified with the approach used in this area of research.

ANSWER: We rewrote section 1 Introduction and better listed the main theoretical and practical implications of using of the intrinsic transfer matrix. The text with the new version of Introduction is :

 “Transfer matrix is an important tool in the study of waves and pulses propagation in finite and infinite homogeneous and inhomogeneous elastic media. The method is widely used in computer simulation or in elastic characterization of different kinds of elastic media. [1],[2],[5]

The main characteristic of matrix methods with respect to other approaches is their simple and compact analytical form and the ease of application in obtaining theoretical results. This has further advantages in the development of numerical methods, where a compact encoding of wave propagation and scattering allows building clear and efficient simulations, which are also easy to test and modify. Computational requirements are also relatively low and the approach is easy to extend. The downside of matrix methods is that while they are well adapted to 1D wave propagation, for larger dimensions they require complicated transfer matrices which are harder to work with and interpret in terms of involved wave phenomena. Nevertheless, in many applications and experimental setups considering a 1D wave propagation is sufficient, which allows an elegant and straightforward approach by a suitable matrix method in order to simulate, explain and characterize the considered system [25].

The elements of the transfer matrix are obtained as a consequence of the boundary conditions and propagation mechanism [9],[10]. For the 1D propagation and solid elastic media, boundary conditions imply continuity of the stress and of the wave function at the interface [3],[4],[12].

Especially in the study of multilayered media the transfer matrix method is used in sound insulation or transmission loss factor, in order to evaluate sound attenuation[8],[13],[14], in automotive design of the interior of the car[15] or the design of multiply connected mufflers[7],[16]. A large volume of research and studies refers to sonic crystals behavior and optimization using the transfer matrix method[18-20]. By using the transfer matrix formalism it is possible to model the behavior of elastic media with inhomogeneities. In that case, transfer matrix approach combines with finite element method in order to describe homogeneous and non-homogeneous acoustic absorbing materials. The characterized acoustic materials are mainly metamaterials made of multiple layers, where at least one layer consists of a non-homogeneous material. The equivalent transfer matrix of the non-homogeneous material is determined and by using the equivalent transfer matrix of the non-homogeneous material coupled analytically in series with other transfer matrices, complex multilayer systems can be modeled easily and quickly in configurations wherein the use of finite element calculation alone will be more expensive and time consumming [11]. Different other methods have been developed to numerically simulate waves in complex materials, some of which take advantage of matrix formulations [26][27]. Especially in the case of polycrystalline materials, velocity surfaces [28] or slowness surfaces [29][30] are used to characterize local anisotropy. Taking into account inhomogeneity and anisotropy  is important in sound wave propagation [31][32] as it gives a clearer view on wave propagation and scattering and also on the emerging sound dispersion in composite or multilayered structures.

Intrinsic transfer matrix represents a special kind of transfer matrix written for amplitudes of the Fourier components of the waves confined in an embedded elastical system. Combining the properties of the intrinsic transfer matrix with a corresponding modal analysis, we can determine some elastic constants of the system. This application of the method is presented in section 2. By imposing the condition that at resonance the eigenvalues of the intrinsic transfer matrix are real, we can generate another form of the resonance condition for the considered elastic system. This extension is presented in section 3.1. The resonance condition may be generalized if we consider the attenuation of the amplitudes of the involved waves. The obtained results in case of attenuation were compared with those obtained by using a simulation based on the model of the coupled oscillators (this method of simulation is very often used for study of multilayered materials). This is presented in section 3.2.  The intrinsic transfer matrix was also applied to study by computer simulation the behavior of a multilayered medium with inhomogeneities, by considering the case in which one layer consists of a nonhomogeneous material. This is presented in section 3.3. One possible effect of inhomogeneity is the appearance of dispersion, i.e. frequency-dependent sound velocity. This is applied in the case of multilayered media, and also for polycrystalline materials composed of anisotropic grains such as metals, or for metamaterials in a larger sense. The simulations are presented in section 3.3. In section 3.4 we approach an optimization procedure for the design of a multilayered medium, where the purpose is to place the sample of interest such as to minimize the sound velocity errors due to frequency determination. The case of a ternary elastic system is being analysed, but the study may be generalized to other complex multilayered media.”

  • The experimental section and the valid reference must be quoted with all the test specifications, conditions, and standards.

 ANSWER: The experimental results obtained by measurements using the intrinsic transfer matrix method correlate very well with the results obtained on similar materials mentioned in References.

  • Material orientation is essential, and there are plenty of known reasons and equation conditions available.

ANSWER: In INTRODUCTION we mentioned a new paragraph containing the description of the advantage and the limitations of application of the transfer matrix method:

                The main characteristic of matrix methods with respect to other approaches is their simple and compact analytical form and the ease of application in obtaining theoretical results. This has further advantages in the development of numerical methods, where a compact encoding of wave propagation and scattering allows building clear and efficient simulations, which are also easy to test and modify. Computational requirements are also relatively low and the approach is easy to extend. The downside of matrix methods is that while they are well adapted to 1D wave propagation, for larger dimensions they require complicated transfer matrices which are harder to work with and interpret in terms of involved wave phenomena. Nevertheless, in many applications and experimental setups considering a 1D wave propagation is sufficient, which allows an elegant and straightforward approach by a suitable matrix method in order to simulate, explain and characterize the considered system [25].

  • The influence of in-homogeneity details was not discussed in the existing literature.

ANSWER: We mentioned this topic in an introductory paragraph and mentioned appropriate bibliographic references:

 “By using the transfer matrix formalism is possible to model the behavior of elastic media with inhomogeneities. In that case, transfer matrix approach combines with finite element method  in order to describe  homogeneous and non-homogeneous acoustic absorbing materials. The characterized acoustic materials are mainly metamaterials made of multiple layers, where at least one layer consists of a non-homogeneous material. The equivalent transfer matrix of the non-homogeneous material is determined and by using the equivalent transfer matrix of the non-homogeneous material coupled analytically in series with other transfer matrices, complex multilayered systems can be modeled easily and quickly in configurations wherein the use of finite element calculation alone will be more expensive and time consuming [11]. Different other methods have been developed to numerically simulate waves in complex materials, some of which take advantage of matrix formulations [26][27]”.

  • Some of the references used in the studies are very old. Kindly include the new articles published in the last 5 years.

ANSWER: We have added the following titles to References:

  1. Pierre-David Létourneau, Ying Wu, George Papanicolaou, Josselin Garnier, Eric Darve, A numerical study of super-resolution through fast 3D wideband algorithm for scattering in highly-heterogeneous media, Wave Motion 2017, 70, 113-134, https://doi.org/10.1016/j.wavemoti.2016.08.012.
  2. Yuriy N. Belyayev, Method for calculating multiwave scattering by layered anisotropic media, Wave Motion 2020, 99, 102664, https://doi.org/10.1016/j.wavemoti.2020.102664
  3. Paul Dryburgh, Wenqi Li, Don Pieris, Rafael Fuentes-Domínguez, Rikesh Patel, Richard J. Smith, Matt Clark, Measurement of the single crystal elasticity matrix of polycrystalline materials, Acta Materialia, 2021, 117551, https://doi.org/10.1016/j.actamat.2021.117551
  4. Yves-Patrick Pellegrini, Causal Stroh formalism for uniformly-moving dislocations in anisotropic media: Somigliana dislocations and Mach cones, Wave Motion 2017,68,128-148, https://doi.org/10.1016/j.wavemoti.2016.09.006.
  5. Harriet Grigg, Barry J. Gallacher, Nathan P. Craig, Robust, high-resolution, indexed 3D slowness surfaces for Rayleigh-type waves on Lithium Niobate via parallelised Newtonian flow phase tracking, Journal of Sound and Vibration 2022, 518, 116533, https://doi.org/10.1016/j.jsv.2021.116533.
  6. Leandro Maio, Paul Fromme, On ultrasound propagation in composite laminates: advances in numerical simulation, Progress in Aerospace Sciences 2022,129,100791, https://doi.org/10.1016/j.paerosci.2021.100791
  7. Radosław Drelich, Mariusz Kaczmarek, Bogdan Piwakowski, Accuracy of parameter identification using the dispersion of surface waves and the role of data quality for inhomogeneous concrete, NDT & E International 2018, 98,195-207, https://doi.org/10.1016/j.ndteint.2018.05.002.

Round 2

Reviewer 1 Report

The authors have addressed all the comments.

Author Response

English was checked and corrected by the staff of the Department of Foreign Languages at Transilvania University Brasov. A pdf version of the paper is attached.

Reviewer 2 Report

I appreciate the authors feedback and the comments which are incoporated in the article. However, I request the authors to read the manuscript throughly and cite the related references. In the present manuscript, the authors cited the article randomly i.e 1, 2, 5 and moved to 25 and jumped to references 9,10.

Please change the references as per the journal format.

Author Response

English was checked and corrected again by the staff of the Department of Foreign Languages at Transilvania University Brasov. The citation of the bibliography has been corrected. A copy of the valid paper is attached.

This manuscript is a resubmission of an earlier submission. The following is a list of the peer review reports and author responses from that submission.

Round 1

Reviewer 1 Report

See my comments in the attached file. Thank you.

Reviewer 2 Report

The manuscript  concerns theoretical investigation of layered elastic materials and its comparison with experiments.

I see interactions with the previous papers of the same authors which are not cited, for instance,

https://www.sciencedirect.com/science/article/abs/pii/S0165212516300166#!

https://www.sciencedirect.com/science/article/abs/pii/S0003682X15000523

The presented theory is not new and the theoretical part repeats the known theory.

The experimental part is too short and poor. It is not clear how it is related to the experiments presented in the previous papers.

I think that the present manuscript is based on the already published results of the same authors. It doesn’t contain essentially new results. Therefore, I do not recommend to publish this paper.